# Evaluating the Performance of Cocopeat and Volcanic Tuff in Soilless Cultivation of Roses

**DOI:** 10.3390/plants13162293

**Published:** 2024-08-17

**Authors:** Malik G. Al-Ajlouni, Yahia A. Othman, Nour S. Abu-Shanab, Lujain F. Alzyoud

**Affiliations:** Department of Horticulture and Crop Science, University of Jordan, Amman 11942, Jordan; ya.othman@ju.edu.jo (Y.A.O.);

**Keywords:** particle size, cut flower, substrate depth, cut flowers, flower quality

## Abstract

Roses are increasingly being grown in soilless systems to increase productivity and reduce the challenges associated with soil-based cultivation. This study investigates the effects of using cocopeat and volcanic tuff substrates, the particle size of tuff, and substrate depth on the growth and flower quality of roses (*Rosa hybrida* L. cv. top secret) grown under greenhouse conditions. The treatments were cocopeat, tuff, cocopeat–tuff mixture, and tuff of particle size of 2 to 4 mm, 0 to 4 mm, and 0 to 8 mm at depths of 20 cm and 40 cm. The results showed that cocopeat had the highest water-holding capacity and photosynthetic rate. Tuff substrates had higher chlorophyll content throughout the growing season. Although flower numbers per plant in cocopeat and tuff from 0 to 8 mm at a depth of 20 cm were statistically similar, tuff from 0 to 8 mm had longer flowering stems and larger post-harvest flower diameters than cocopeat. An increase in the tuff depth from 20 to 40 cm decreased the flower number and main stem diameter. In conclusion, while cocopeat promotes rapid initial growth, volcanic tuff substrates, particularly tuff from 0 to 8 mm at a depth of 20 cm, provide long-term benefits for flower quality and plant health.

## 1. Introduction

Roses belong to the Rosaceae family, the largest and most significant horticultural family globally in terms of economic impact [1]. Historical records show that roses were cultivated in Europe and China as early as 5000 years ago to fulfill utilitarian requirements like making oils and medications [2,3]. Due to their exceptional beauty, roses have been portrayed in art and sculpture for centuries as well as frequently mentioned in poetry and religious texts [4]. Growing roses as cut flowers is a very profitable business because the yield is twice as high as the cost [5]. In the United States, sales in 2014 brought in USD 204 million, with 1808 growers producing 36 million plants worth [6]. According to The International Association of Horticultural Producers (AIPH), the total number of roses exported to European Union countries in 2020 was about 8.06 billion flowers, contributing to a total value of USD 1.4 billion [7].

Soilless culture is any method of growing plants without using soil as a rooting medium, in which the inorganic nutrients absorbed by the roots are supplied via irrigation water [8]. Soilless culture is an alternative and more intensive method to conventional soil cultivation [9]. The cost, local availability, and experience in substrate (growing medium) use are usually the determining factors for choosing a particular substrate type. Among the various soilless culture practices, the use of substrate is the easiest to be adopted by growers [10]. As growing media, various substrates including gravel, peat, sand, tuff, perlite, coir, and rockwool have been used in soilless culture systems, either alone or in combination [11,12].

There are many different types of soilless substrates used in cut roses including zeolite, perlite, cocopeat, and volcanic tuff [13,14]. However, selecting a growing substrate for optimal growth and productivity in soilless cultivation can be challenging [15]. In order to achieve successful shoot growth and root penetration, it is essential to possess certain characteristics. These include a high water-holding capacity (WHC), total porosity, and low bulk density. These factors play an important role in providing the necessary conditions for plants to establish and thrive in a soilless environment [16,17]. Growing substrates such as peat, compost, and cocopeat are among the organic substrate sources, whereas sand, perlite, vermiculite, gravel, pumice, expandable clay, and volcanic tuff are categorized as inorganic substrate materials [18].

Cocopeat is gaining popularity as a growing substrate. Cocopeat is a byproduct of the processing of coconut husks, often known as coir dust, cocopeat, coir pith, or just coir. It is a versatile natural fiber derived from the mesocarp tissue or husk of the coconut. The husk contains 20–30% fiber of various lengths and can store eight to nine times its weight in water [19]. Cocopeat is known for its moisture retention and fibrous structure, which promotes aeration, root development, and oxygen uptake [20]. Therefore, cocopeat becomes an optimal medium for plant root development [21]. In addition, cocopeat is considered a renewable resource and environmentally friendly [22].

Volcanic tuff is a popular soilless substrate in the Mediterranean region. It is used to grow cut flower species such as roses and lily. Volcanic tuffs are pyroclastic rocks that are porous and generally soft. They are typically created through the lithification and cementation of volcanic ash or dust [23]. Tuff has excellent aeration, cation exchange capacity, and resistance to acidity [24,25]. The particle size of commercial tuff for soilless agriculture is between 1 and 10 mm. It is important to note that the diameter of these particles plays a significant role in determining the availability of water and aeration within the substrate. Therefore, it is essential to establish the suitable particle size for the successful growth of plants in a soilless system utilizing volcanic tuff [26]. Local cut flower growers commonly use tuff from 0 to 8 mm for rose plants and tuff from 0 to 4 mm for lily plants in a 20 cm depth soilless culture system. Although both cocopeat (organic substrate) and tuff (mineral source) have been recommended as an environmentally friendly substrate for soilless culture, their potential effect on the growth and productivity of cut flowers is not fully understood. In addition, research studies that focus on the effect of particle size (diameter) and substrate depth on flower postharvest quality (specifically rose flowers) are limited.

The objectives of this study were to explore the effects of cocopeat and volcanic tuff substrates, tuff substrate size, and depth of volcanic tuff substrate on the growth and flower quality of rose plants grown in a soilless system under greenhouse conditions. In this research study, we investigated how different types and characteristics of substrates affected various growth parameters, such as root, leaf, and stem components in roses. The findings of this study are expected to provide valuable information for commercial cut flower growers especially those looking to optimize soilless systems; in order to improve flower quality and increase yield under controlled greenhouse conditions.

## 2. Results

### 2.1. Physical and Chemical Properties of Substrates

The physical and chemical properties of substrates that were tested at the beginning of the experiment are presented in Table 1. Cocopeat had the highest WHC (66.2%) and tuff from 2 to 4 mm had the lowest (21.5%). Mixing tuff (0 to 4 mm) with cocopeat increased WHC by 30% compared to the tuff from the 0 to 4 mm substrate. An increase in the particle size of the tuff decreased WHC. An increase in the particle size of the tuff increased the air-filled porosity. Tuff from 0 to 8 mm had the highest air-filled porosity. Interestingly, air-filled porosity decreased in the cocopeat + tuff mixture by 31% than the tuff from the 0 to 4 mm substrate. In terms of bulk density, cocopeat and tuff from 0 to 4 mm had the lowest and highest values, respectively. Cocopeat had the lowest pH of all tuff substrates. Volcanic tuff material had lower electrical conductivity (EC) than cocopeat.

### 2.2. Plant Growth and Flower Quality

Cocopeat maintained a high photosynthesis rate in April, high stomatal conductance in April and August, and a high transpiration rate in April and August (Table 2). In October, photosynthesis and transpiration rates were statistically similar among all treatments, while stomatal conductance in tuff from 2 to 4 mm, tuff from 0 to 8 mm (20 cm height), tuff from 0 to 8 mm (40 cm height), and cocopeat were statistically similar. In addition, there was no significant photosynthesis in August.

Although the number of flowers per plant in cocopeat, tuff from 0 to 4 mm, and tuff from 0 to 8 mm (20 cm height) were statistically similar, the number of flowers per plant were higher in cocopeat by 19% and 9% than tuff from 0 to 4 mm and tuff from 0 to 8 mm (20 cm height), respectively (Table 3).

The cocopeat + tuff (0 to 4 mm) mixture substrate had fewer flowers per plant compared to cocopeat, tuff from 0 to 4 mm, and tuff from 0 to 8 mm (20 cm height). An increase in the tuff substrate depth from 20 cm to 40 cm reduced the flower number by 16%. Interestingly, all tuff substrates maintained higher flowering stem lengths as well as flowering lengths than the cocopeat substrate. There were no significant differences in the flowering stem fresh weight, flowering stem diameter, vase life, and flower diameter across treatments. Although the flower diameter at harvesting was statistically similar across treatments, the flower diameter varied on the fifth day in the vase.

Tuff from 2 to 4 mm, tuff from 0 to 8 mm (20 cm height), and tuff from 0 to 8 mm (40 cm height) had larger flower diameters five days after harvesting in the vase compared to the cocopeat and cocopeat + tuff mixture. Five days after harvesting, neither substrate depth of the tuff from 0 to 8 mm improved the flower diameter.

Cocopeat, tuff from 0 to 8 mm (20 cm depth), and tuff from 0 to 4 mm had the thickest main stem diameters (Table 4). The 0 to 8 mm tuff substrate with a 40 cm depth reduced the main stem diameter by 21% compared to the 20 cm depth. In terms of roots, cocopeat had longer roots compared to other treatments at a depth of 20 cm. Root dry weight was the lowest in the tuff from 2 to 4 mm. An increase in the depth of the 0 to 8 mm tuff treatment from 20 to 40 cm increased the root length by 44%. Across tuff treatments, chlorophyll content was higher than the cocopeat substrate (Figure 1).

## 3. Discussion

### 3.1. Substrate Physical and Chemical Properties

The results of the research provide insight into the relation between the physical and chemical characteristics of the substrate and its impact on plant physiology as well as the flower quality of the rose. In this study, cocopeat had the highest WHC. A’saf et al. [27] found that cocopeat had the greatest WHC compared to the tuff (0 to 2, 2 to 4, 0 to 4, and 4 to 8 mm), sand, soil, LECA, peat moss, and perlite. Because of its fibrous structure, which effectively retains moisture, cocopeat has become a substrate with a significant WHC [28]. Organic matter is essential for enhancing the mixture’s ability to hold water as well as the health of the growth media, CO_2_ respiration, microbial abundance, and activity [29]. Considering that cocopeat had the highest WHC (62.2%) compared to the tuff (21.5%), the combined cocopeat and tuff mixture had higher WHC and lower air-filled porosity when compared to the tuff substrate solely. In terms of particle size, Brady and Weil [30] concluded that fine particles typically had larger total surface areas than coarse particles and held more water and nutrients. This explained why tuff from 0 to 4 mm had greater WHC than tuff from 2 to 4 mm and tuff from 0 to 8 mm. Excess water and fine particles reduced air-filled porosity, which reflects microbial activity and root health [16]. In the current study, air-filled porosity showed a positive relation with the particle size of tuff. This was consistent with the results of Wallach et al. [31], who found that the fine particle size increased bulk density and decreased porosity. Higher air-filled porosity was found in coarse particles [32,33] but too coarse a substrate could impede plant growth [34]. Conversely, fine particle substrates held more water but also made plants more vulnerable to nutrient deficiencies and root diseases [35,36]. These insights provide valuable guidance for growers on how to choose substrate compositions that maximize nutrient availability, aeration, and water retention. This will improve crop productivity in soilless cultivation systems.

Moreover, variations in EC and pH substrates were observed between the various substrates (Table 1). Tuff substrates had higher EC values and lower pH values than cocopeat. These variations can be ascribed to changes in the mineral content and composition of the substrate, highlighting the significance of managing pH and EC to provide the best possible growing conditions for plants [12]. We used a fertigation solution to bring the pH down to 5.5–6.5 in order to overcome the pH variation in the studied substrates.

### 3.2. Plant Growth and Flower Quality

Cocopeat demonstrated the highest photosynthesis and stomatal conductance in April, as well as high transpiration rates in both April and August. This suggests that cocopeat initially provided ideal necessary growth conditions for root and shoot activity, promoting efficient gas exchange, and water movement [37]. By October, tuff from 0 to 8 mm had surpassed cocopeat in stomatal conductance, while photosynthesis and transpiration showed no significant differences between the treatments. By the end of the experiment, we noticed that the cocopeat had dropped in height by five cm (indicating media decomposition) while tuff substrates sustained the same height. According to Krishnapillai et al. [21], coconut husks (cocopeat raw materials) begin to decompose in about 2 to 3 months under wet conditions. In contrast, tuff media maintained a consistent structure, ensuring stable aeration and avoiding the issues of compaction and poor root aeration associated with fine particles [16,38]. It is preferable to use a slow decomposition material when selecting a media for soilless culture to reduce the changing costs of substrates each year [9]. Volcanic tuff can last for many years due to its stability and resistance to solubility or degradation [39]. This stability might partially contribute to the stomatal conductance increment in October. As a result, while cocopeat promotes rapid initial growth, decomposition reduces its long-term effectiveness. Therefore, the substrate with improved aeration and structure promotes more sustainable long-term growth.

The flower size, number of flowers per stem, plant height, and longevity are key elements of high-quality flowers in the cut-flowers market [40,41]. Gizas and Savvas [33] studied different pumice particle grades (0 to 2, 0 to 5, 0 to 8, or 4 to 8 mm) on the flower yield of roses. They found that the number of flowers per plant, flower length, and flower weight were not influenced by the particle size, except for 4 to 8 mm. In the current study, the tuff diameter ranged from 2 to 4 mm and had fewer flowers per plant compared to other treatments. However, tuff from 0 to 8 mm (20 cm depth) maintained high values of flower numbers per plant, flower stem length, and flower diameter after five days in the vase. Although cocopeat had a high number of flowers per plant, it had shorter flowering stems and smaller flower diameters. Therefore, tuff from 0 to 8 mm (20 cm height) retained high flower quality. In particular, it had high porosity and low EC values compared to the cocopeat substrate. Li et al. [42] found that an increase in the total porosity of the substrate markedly affected plant morphology, growth, and biomass [42]. In addition, an increase in air porosity accelerated flower emergence and blooming [43]. The particle size and heterogeneity of growing media, as discussed by Huang et al. [44], are likely to contribute to improved aeration and water retention, resulting in increased flower growth and longevity.

Interestingly, an increase in the depth of the tuff substrate from 20 cm to 40 cm reduced the flower number by 16%, implying that deeper substrates may create unfavorable conditions for flower growth. This finding could be attributed to substrate compaction or reduced aeration at greater depths. Air diffusion is less effective in the deeper part of the soil and in determining soil aeration [45].

Although this study showed that the flower diameter at harvest time was statistically similar across all treatments, differences emerged five days later in the vase. Tuff substrates (2 to 4 mm, 0 to 8 mm at 20 cm and 40 cm heights) had larger flower diameters than cocopeat and cocopeat–tuff mixtures. This underscores the impact of the cocopeat substrate type on post-harvest flower quality while tuff substrates promote better flower preservation. Tuff includes essential micronutrient elements like iron and magnesium [46]. Iron and magnesium influence the chlorophyll content, which is crucial for plant growth and flower development [47].

In terms of plant morphology, cocopeat promoted longer root lengths at a 20 cm depth. Doubling the substrate depth in the 0 to 8 mm tuff treatment (20 to 40 cm) increased the root length by 44%. Although tuff from 0 to 8 mm (40 cm) increased the root length, it reduced the main stem diameter by 21%. Root dry weight was the lowest in tuff from 2 to 4 mm than other treatments, where tuff from 2 to 4 mm had the lowest WHC. According to Deepagoda et al. [16], low substrate moisture content causes a water deficit, prevents nutrient uptake, and slows plant growth.

Although fertigation solution was used to fertigate plants daily, variations in the chemical composition between volcanic tuff cannot be ignored. According to Almjadleh et al. [46], the chemical composition values for Jordanian volcanic tuff were 686 mmol/L SiO_2_, 122 mmol/L Al_2_O_3_, 98 mmol/L Fe_2_O_3_, 4 mmol/L MnO, 36 mmol/L TiO_2_, 129 mmol/L CaO, 21 mmol/L K_2_O, 4 mmol/L P_2_O_5_, 194 mmol/L MgO, and 44 mmol/L Na_2_O, while the chemical composition values of cocopeat were 3.1 mmol/L N, 0.12 mmol/L P, 4 mmol/L K, 0.14 mmol/L Ca, 0.23 mmol/L Mg, and 1.3 mmol/L S [48]. Hence, volcanic tuff is rich in minerals like iron, silicon, and magnesium, which influence chlorophyll production in the leaves [49,50]. Shaahan et al. [51] found significant positive correlations between the readings from the meter and the magnesium concentrations in the leaves of grapevine, guava, and mango. Therefore, tuff substrates consistently maintained higher chlorophyll content compared to cocopeat, as reflected in overall plant health and vitality. In contrast, chlorophyll content measurements in cocopeat showed a dramatic decline in the September reading, indicating deteriorating plant health. Moreover, the color yellow was noticed on rose leaves that were grown in the cocopeat substrate during the latter months of the experiment. Concurrently, gas exchange variable readings of roses in cocopeat were degraded from giving the best results in April to becoming statistically similar or even worse than the tuff substrates in the recent measurements. This is another indicator that cocopeat does not sustain plant health as effectively as tuff. Therefore, using the tuff substrate is more sustainable than a cocopeat substrate. According to Silber and Raviv [39], volcanic tuff can last for many years due to its stability and resistance to solubility or degradation.

## 4. Materials and Methods

### 4.1. Experiment Layout and Plant Material

The study was conducted at the University of Jordan, Amman (lat. 32°0′40.4316″ N, long. 35°52′20.3628″ E) between October 2022 and December 2023. The experiment was carried out in a 10 × 30 m glass greenhouse. The greenhouse was equipped with a fan and pad system that kept the temperature stable during the hot period. The weather data inside the greenhouse during the experiment were recorded by the Hobo Pendant temp/light data logger (Onset Computer Corp., Bourne, MA, USA) (Figure 2). A closed soilless system was used to grow rose plants. Rooted rose cuttings (*Rosa hybrida* L. cv. Top Secret) were planted on 1 October 2022. During the first period of the experiment, rose plants were regularly pruned by pinching and bending down all weak stems to maximize the photosynthetic leaf area. Rose plants were trained and prepared for harvesting according to the procedure of Dole and Wilkins [52]. Harvesting of rose flowers started in April 2023 and continued until 31 December 2023, when the experiment ended. Three rose plants were transplanted to each planter (0.42 m wide and 1 m long). We built four linear containers (7 m long) that were made of galvanized steel sheets (Figure 3 and Figure 4). The galvanized steel sheets were bent in U-shapes to hold the growing substrates and the drainage layer. A heavy-duty polyethylene sheet (400 microns) was placed over the interior base of each linear container as a waterproof layer. A 10 cm-thick layer of solid gravel (particle size 10 to 14 mm) was placed in the base of the linear container as a drainage layer. The north part of the container was raised to create a 1% slope that towered the drainage hole. Each container had a drain hole that led to a drainage hose to drain leachate water, which was collected in a 100 L underground tank. Then, the drained water was pumped to the fertigation tank. A galvanized steel barrier was used to divide each linear container into six separate planters (experimental unit). Above the drainage layer, a nonwoven geotextile membrane (100 microns) was placed all over the planter sides to hold the growing substrates. The dimensions of the planter were 0.42 m wide and 1.00 m long. Five substrates (treatments) had a depth of 20 cm and only one substrate had a depth of 40 cm. Each planter represented a specific substrate (treatment). Hence, six substrate treatments were performed in this experiment. These six substrates were as follows: tuff from 2 to 4 mm, tuff from 0 to 4 mm, tuff from 0 to 8 mm (20 cm substrate depth), tuff from 0 to 8 mm (40 cm substrate depth), cocopeat + tuff from 0 to 4 mm (1:1 *v*/*v*), and cocopeat. The distribution of particle sizes used in the experimental treatments for tuff from 2 to 4 mm, tuff from 0 to 4 mm, and tuff from 0 to 8 mm is shown in Figure 5.

Each plant was irrigated daily using a drip irrigation system. Irrigation was scheduled by using an irrigation controller (Hunter X-Core^®^, Hunter Industries, San Marcos, CA, USA). The irrigation was scheduled to run for 8 min once a day (at 8:00 a.m.) from October to March and twice a day (at 8:00 a.m. and 4:00 p.m.) from March to October. One emitter per plant was used with a flow rate of 8 L per hour. Rose plants were fertigated through the irrigation schedule. A complete liquid fertilizer (Gold Leaf, Planterbio^®^, Derbyshire, UK) was added to the irrigation tank to ensure proper nutrient levels for strong growth and development. The composition values of the fertilizer were 642 mmol/L nitrogen, 35 mmol/L P_2_O_5_, 170 mmol/L K_2_O, 38 mmol/L calcium, 49 mmol/L Sulphur, 9 mmol/L magnesium, 3 mmol/L iron, 0.3 mmol/L zinc, 0.2 mmol/L manganese, 1 mmol/L boron, 0.02 mmol/L copper, and 0.01 mmol/L molybdenum. The fertigation solution had 5.5 pH and the EC was less than 1.5 dS·m^−1^. The fertigation tank was emptied and cleaned before being filled with water and fertilizer every week. Water-holding capacity (WHC), bulk density, air-filled porosity, EC, and pH for the growing substrates were measured at the beginning of the experiment according to the procedure of Brady and Weil [30]. The electrical conductivity meter (YK-22CT, Lutron Electronic Enterprise CO, Taipei, Taiwan), and pH meter (pp-203 pH, GOnDO Electronic Co., Taipei, Taiwan) were measured at the beginning of the experiment for all growing substrates (Table 1).

### 4.2. Plant Growth and Flower Quality

Chlorophyll content was measured in the middle of the plant leaf by a chlorophyll concentration meter (MC-100 Chlorophyll Meter, Apogee Instruments, UT, USA). This measurement was taken three times during the experiment between 10:00 a.m. and 12:00 p.m. (Figure 1). Leaf gas exchange measurements were recorded three times throughout the experiment (Table 2). Photosynthesis rate, stomatal conductance, and transpiration were measured using a portable photosynthesis system (LI-6400XT, LICOR, Lincoln, NE, USA). The uniform leaf was sampled and placed in the gas chamber in the middle of the leaf from each plant on 4 June, 21 August, and 17 October 2023.

The number of flowers per plant, flowering stem length, flowering stem fresh weight, flowering stem diameter, vase life, flower diameter at harvesting, and flower diameter measurements after five days in the vase were taken throughout the experiment. The flower diameter after five days in the vase was the measurement of the flower’s widest point five days after it had been cut and placed in a vase. At the end of the experiment, the root length, root dry weight, and main stem diameter (5 cm above the grafting point) were recorded.

### 4.3. Statistical Analysis

A randomized complete block design with six treatments (different substrates) was replicated four times. Data were analyzed using SAS statistical software (Version 9.4 for Windows; SAS Institute, Cary, NC, USA). The significant level was defined at *p* ≤ 0.05. For each experimental unit, the measurements of three rose plants were averaged before running the statistical analysis. Substrate properties, leaf gas exchange, flower quality, main stem diameter, root length, and root dry weight were statistically analyzed using the analysis of variance (ANOVA), and the means were separated by Tukey’s honestly significant difference (HSD) test.

## 5. Conclusions

This study examines the effect of various substrates, specifically cocopeat and various particle sizes of tuff on the physical and chemical properties of growing media and their subsequent effects on plant growth and flower quality. The findings show that cocopeat—with its high WHC—promotes early robust growth due to efficient moisture retention and favorable conditions for gas exchange and water movement. However, cocopeat’s tendency to decompose over time reduces aeration and decreases long-term plant health and productivity. Conversely, tuff substrates, particularly tuff with a particle size of 0 to 8 mm, provided superior structural stability and aeration, which are essential for sustaining plant growth over longer periods. Although the flower numbers per plant in cocopeat and tuff from 0 to 8 mm (20 cm depth) were statistically similar, tuff from 0 to 8 mm had longer flowering stems and larger flower diameters after harvest than cocopeat. Therefore, tuff from 0 to 8 mm maintained higher flower quality.

This research also underscores the importance of substrate depth, revealing that deeper tuff substrates can negatively influence flower numbers and stems. This finding highlights the need for optimizing substrate depth to balance aeration and root growth. Overall, while cocopeat offers advantages for early plant development, tuff substrates, particularly tuff substrates from 0 to 8 mm, provide more sustainable benefits for long-term growth and flower quality. The stability and nutrient-rich composition of tuff, including essential elements like iron and magnesium, enhance chlorophyll content and overall plant vigor. Finally, we recommend tuff from 0 to 8 mm (20 cm depth) for its superior balance of aeration, moisture, structural stability, and nutrient availability, ensuring long-term plant health and flower quality of rose plants grown in soilless cultivation systems.

## Figures and Tables

**Figure 1 plants-13-02293-f001:**
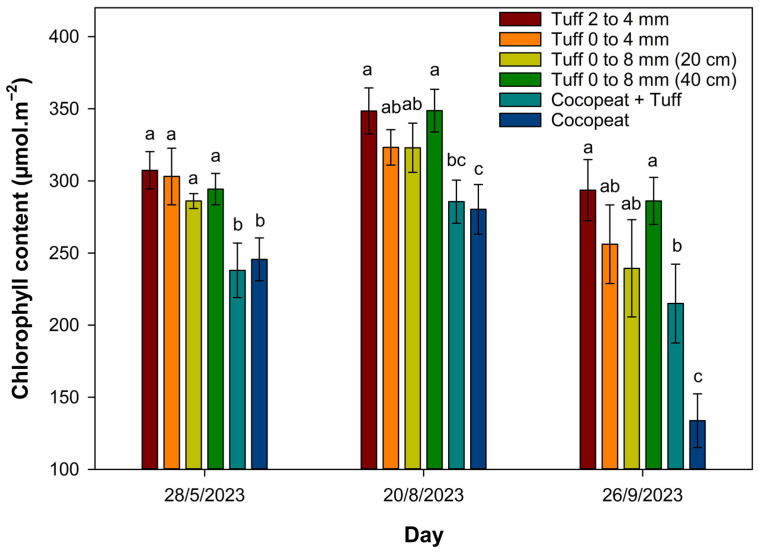
Chlorophyll content of rose plants grown in different soilless substrate mixtures. Measurements were determined on the youngest fully mature leaves on 28 May, 20 August, and 26 September 2023; bars represent the mean for different soilless substrates; the same letter within each group (date) is not significantly different according to Tukey‘s honestly significant difference (HSD) test (*p* ≤ 0.05). The error bars show the standard error of the mean (*n* = 4).

**Figure 2 plants-13-02293-f002:**
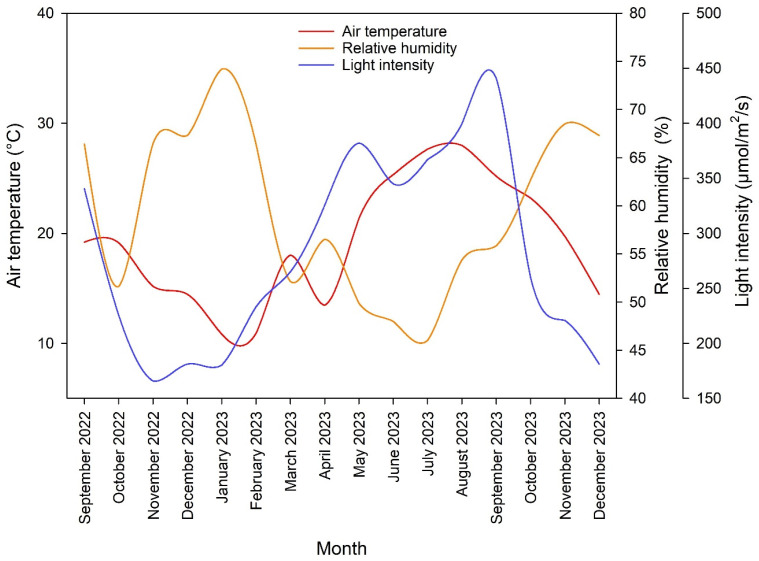
Weather data inside the greenhouse between September 2022 and December 2023.

**Figure 3 plants-13-02293-f003:**
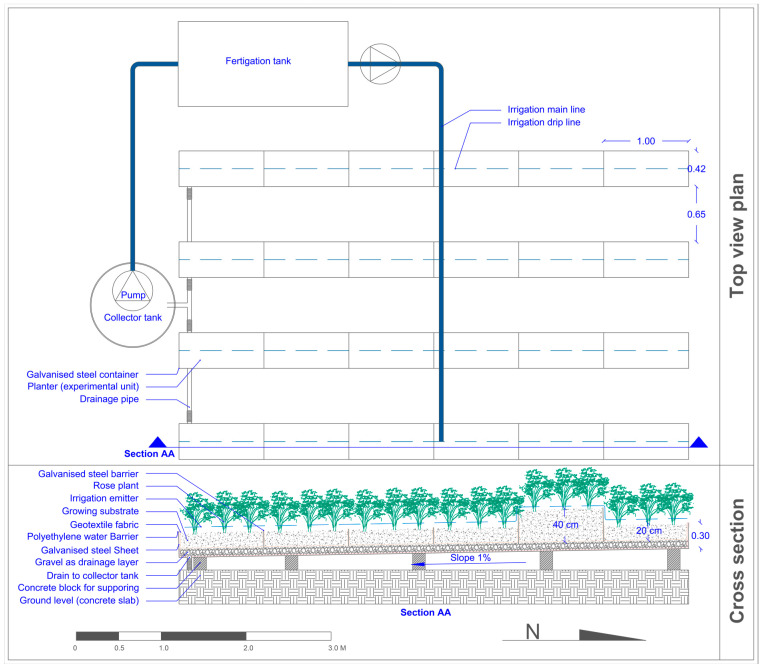
Experiment layout plan illustrating the top view and cross-section of rose plants cultivated in soilless culture substrates in the greenhouse setup using closed soilless systems. The top view details the arrangement and spacing of four planting containers, where each container includes six planters. Each planter includes three rose plants. The cross-section provides insights into the structure and composition of the soilless media layers.

**Figure 4 plants-13-02293-f004:**
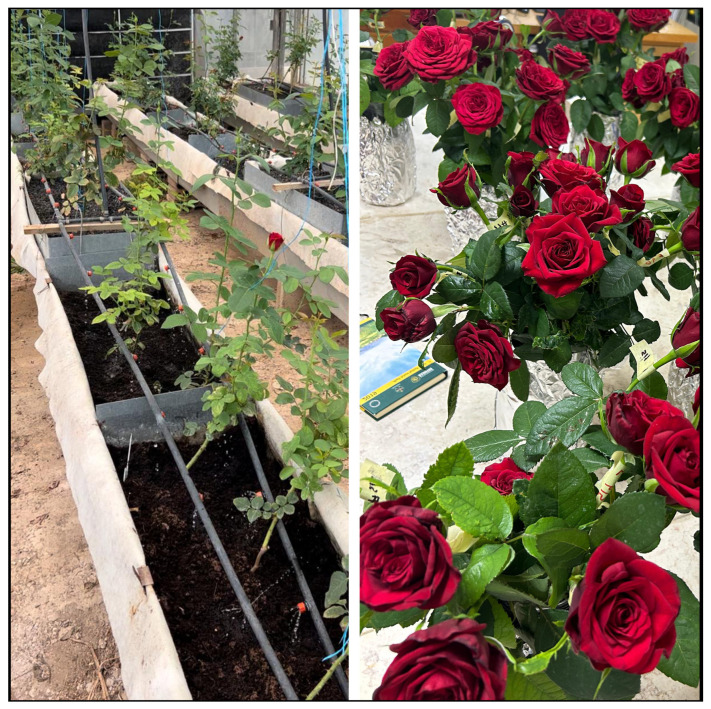
Rose plants were cultivated in soilless culture substrates within the greenhouse environment (**left**). These roses were then transported to a laboratory setting and placed in a vase for vase life measurement (**right**).

**Figure 5 plants-13-02293-f005:**
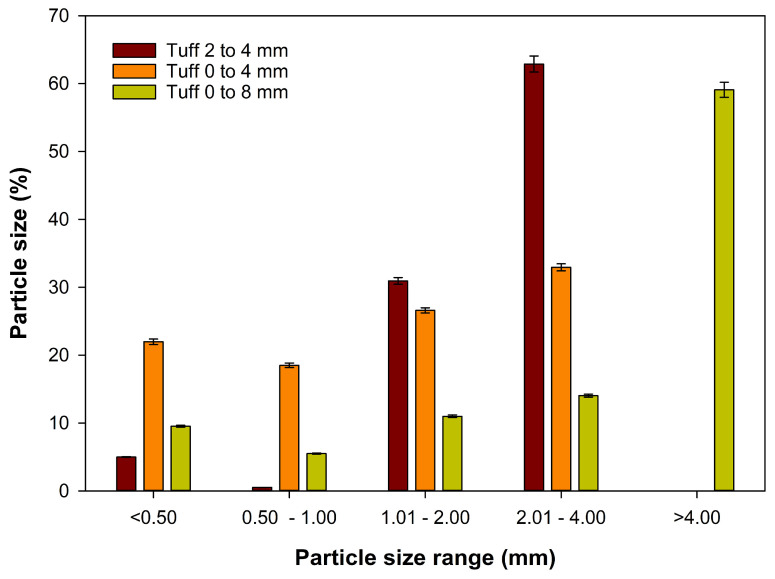
The particle size distribution of three types of volcanic tuff (varying by particle size range). The error bars show the standard error of the mean (*n* = 3).

**Table 1 plants-13-02293-t001:** Physical and chemical properties of substrates (*n* = 4).

Growing Substrate	Water-Holding Capacity (%)	Air-Filled Porosity (%)	Bulk Density (g·cm^−3^)	pH	EC (dS·m^−1^)
Tuff from 2 to 4 mm	21.54 e *	51.98 b	0.70 b	7.16 a	0.241 d
Tuff from 0 to 4 mm	37.28 c	29.89 d	0.88 a	7.09 a	0.383 c
Tuff from 0 to 8 mm	28.92 d	61.43 a	0.63 c	7.05 a	0.279 cd
Cocopeat + tuff	48.51 b	28.92 d	0.62 c	6.61 b	0.577 b
Cocopeat	66.24 a	41.82 c	0.10 d	5.26 c	0.770 a
*p*-value	<0.0001	<0.0001	<0.0001	<0.0001	<0.0001

* Numbers followed by the same letter within a column are not significantly different according to Tukey’s honestly significant difference (HSD) test (*p* ≤ 0.05).

**Table 2 plants-13-02293-t002:** Photosynthesis rate, stomatal conductance, and transpiration for rose plants that were grown in different soilless mixtures between October 2022 and December 2023 (*n* = 4).

Treatments	Photosynthesis (µmol m^−2^·s^−1^)			Stomatal Conductance (mol m^−2^·s^−1^)			Transpiration (mmol m^−2^·s^−1^)		
	April	August	October	April	August	October	April	August	October
Tuff from 2 to 4 mm	39.32 c *	35.21	31.60	0.09 b	0.20 ab	0.11 ab	4.28 b	5.52 ab	2.73
Tuff from 0 to 4 mm	41.31 bc	36.18	31.04	0.10 b	0.16 ab	0.07 b	4.50 b	5.39 ab	2.19
Tuff from 0 to 8 mm (20 cm)	39.65 c	35.26	32.12	0.10 b	0.11 b	0.13 ab	3.29 b	3.71 b	3.21
Tuff from 0 to 8 mm (40 cm)	40.68 c	35.71	34.98	0.10 b	0.15 ab	0.15 a	4.36 b	5.05 ab	3.10
Cocopeat + tuff	42.04 ab	36.61	31.58	0.11 b	0.16 ab	0.09 b	5.55 ab	5.77 ab	2.69
Cocopeat	44.14 a	36.02	31.66	0.18 a	0.25 a	0.10 ab	7.22 a	6.94 a	2.81
*p*-value	0.0003	0.1906	0.1581	0.0002	0.0196	0.0219	0.0003	0.0152	0.2427

* Numbers followed by the same letter within a column are not significantly different according to Tukey’s honestly significant difference (HSD) test (*p* ≤ 0.05).

**Table 3 plants-13-02293-t003:** Flower quality components; flowering stem height, flowering stem fresh weight, flowering stem diameter, number of flowers per plant, flower diameter at harvesting, flower diameter after five days in the vase, and vase life for rose plants that were grown in different soilless mixtures between October 2022 and December 2023 (*n* = 4).

Treatments	Flower Number per Plant	Flowering Stem Length (cm)	Flowering Stem Fresh Weight (g)	Flowering Stem Diameter (mm)	Vase Life (day)	Flower Diameter at Harvesting (mm)	Flower Diameter after Five Days in Vase (mm)
Tuff from 2 to 4 mm	12.25 bc *	39.32 a	32.50	4.26	10.15	26.67	76.52 a
Tuff from 0 to 4 mm	13.08 ab	39.10 a	31.18	4.32	9.87	26.36	71.33 b
Tuff from 0 to 8 mm (20 cm)	14.36 ab	38.91 a	30.69	4.38	10.29	26.51	72.74 ab
Tuff from 0 to 8 mm (40 cm)	12.08 bc	39.60 a	31.48	4.32	10.27	26.16	72.73 ab
Cocopeat + Tuff	10.17 c	38.23 a	28.84	4.16	10.11	25.78	68.18 c
Cocopeat	15.58 a	37.78 b	28.34	4.29	10.02	26.21	68.09 c
*p*-value	0.0039	0.0412	0.127	0.5084	0.8387	0.5796	0.0002

* Numbers followed by the same letter within a column are not significantly different according to Tukey’s honestly significant difference (HSD) test (*p* ≤ 0.05).

**Table 4 plants-13-02293-t004:** Main stem diameter, root length, and root dry weight for rose plants that were grown in different soilless mixtures between October 2022 and December 2023 (*n* = 4).

Treatments	Main Stem Diameter (mm)	Root Length (cm)	Root Dry Weight (g)
Tuff from 2 to 4 mm	8.73 b *	22.77 c	5.66 b
Tuff from 0 to 4 mm	10.29 a	22.43 c	10.05 a
Tuff from 0 to 8 mm (20 cm)	10.39 a	20.10 c	8.72 a
Tuff from 0 to 8 mm (40 cm)	8.58 b	29.00 ab	9.34 a
Cocopeat + tuff	8.80 b	24.43 bc	9.28 a
Cocopeat	10.43 a	34.30 a	8.71 a
*p*-value	0.0077	0.0031	0.0298

* Numbers followed by the same letter within a column are not significantly different according to Tukey’s honestly significant difference (HSD) test (*p* ≤ 0.05).

## Data Availability

Data are contained within the article.

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
