# Peer review of "Evaluating the Performance of Cocopeat and Volcanic Tuff in Soilless Cultivation of Roses"

_plants, 2024, doi:10.3390/plants13162293_

Round 1

Reviewer 1 Report

Comments and Suggestions for Authors   Hello, this is good work and has an important practical application. The introduction is adequate, although it could be more detailed regarding the surface and types used in the production area where the study is carried out. The objectives are clear and well defined. The results are clear and have been presented very accurately both in the tables created and in the graphs used, this helps us a lot to visualize the results obtained clearly and easily. As for the discussion, it is good and has an adequate bibliography for comparison and analysis of results. Regarding the material and methods part, the infrastructure used for the study is well described, especially the location of the plants. The irrigation information is good, however the information they provide regarding the nutrient solution used seems insufficient to me, and it should also have been expressed in mmol/l or meq/l, which is the way in which we understand and analyze it easily. , and this is very important because the drainage obtained during the test was recirculated, so it would have been very important to have also known the chemical values ​​of the drainage. Finally, I think that the greenhouse in which the study was carried out should have been described for the reader's information. Finally, the conclusions are clear, appropriate and respond exactly to the objectives set at the beginning of the study. Icono de Validado por la comunidad    

Author Response

Dear Reviewer,

Thank you for your constructive and thoughtful comments. We have revised the manuscript according to your feedback and suggestions. As detailed below, we have addressed the concerns you raised. We believe these revisions have significantly strengthened the manuscript and hope you will now find it suitable for publication.

Comments 1: The information they provide regarding the nutrient solution used seems insufficient to me, and it should also have been expressed in mmol/l or meq/l, which is the way in which we understand and analyze it easily. , and this is very important because the drainage obtained during the test was recirculated, so it would have been very important to have also known the chemical values ​​of the drainage.

Response 1: We agree. We converted the units from percent to mmol/L

A complete liquid fertilizer (Gold Leaf, Planterbio®, Derbyshire, United Kingdom) was added to the irrigation tank to ensure proper nutrient levels for strong growth and development. The composition of the fertilizer was 642 mmol/L nitrogen, 35 mmol/L P2O5, 170 mmol/L K2O, 38 mmol/L calcium, 49 mmol/L Sulphur, 9 mmol/L magnesium, 3 mmol/L iron, 0.3 mmol/L zinc, 0.2 mmol/L manganese, 1 mmol/L boron, 0.02 mmol/L copper, and 0.01 mmol/L molybdenum.

According to Almjadleh et al. [43], the chemical composition for Jordanian volcanic tuff was 686 mmol/L SiO2, 122 mmol/L Al2O3, 98 mmol/L Fe2O3, 4 mmol/L MnO, 36 mmol/L TiO2, 129 mmol/L CaO, 21 mmol/L K2O, 4 mmol/L P2O5, 194 mmol/L MgO, and 44 mmol/L Na2O, while the chemical composition of cocopeat was 3.1 mmol/L N, 0.12 mmol/L P, 4 mmol/L K, 0.14 mmol/L Ca, 0.23 mmol/L Mg, and 1.3 mmol/L S [45].

Comments 2: I think that the greenhouse in which the study was carried out should have been described for the reader's information.

Response 2: We Agree. Two sentences were added to give more information to the reader.

The experiment was carried out in a 10*30 m glass greenhouse. The greenhouse was equipped with fan and pad system that kept the temperature stable during the hot period.

Reviewer 2 Report

Comments and Suggestions for Authors

This work studied the performance of cocopeat and volcanic tuff in soilless growing media for rose cultivation. This work has practical application value,but need improvement before reconsideration. 

1. More keywords could be added.

2. An additional paragraph about the aim and objectives of this work could be redrafted in the introduction part.. 

3. If possible, please provide the information about the latest data of the economic importance of roses from AIPH in the introduction part..

4. There are many different types of soilless substrate using in cut roses, please give a brief introduction. To my knowledge, cocopeat is also widely used in cut roses, please add some more information in the introduction part.

5. The ratio of tuff and cocopeat could be added at the first look, Line 75-76.

6. Line 128. there is no necessary to compare May and Sep. 

7. Line 152-153, i do not understand how you get the result. Please explain the part clearer.

8. Line180-181, Line185-186. is it a assumption or you proved it?

9. In 2.2, the index of Photosynthetic need to change to photosynthetic rate (Pn).

10. The replicate numbers (n=?) should be added to each table.

11. The meaning of testing the indicator of “flower diameter after five days” needs to be explained, why you choose this indicator for postharvest quality?

Others

Please check the language carefully, for example, Line 190, tough or though?

Figure 3 is not clear, please improve the resolution.

I suggest authors repeat more plants in their future studies to reduce experiment errors.

Comments on the Quality of English Language

The English language quality needs to be improved. Please check once the language  carefully.

Author Response

Dear Reviewer,

Thank you for your constructive and thoughtful comments. We have revised the manuscript according to your feedback and suggestions. As detailed below, we have addressed the concerns you raised. We believe these revisions have significantly strengthened the manuscript and hope you will now find it suitable for publication.

Comments 1:   More keywords could be added.

Response 1: We agree. two more keywords were added.

Comments 2:  An additional paragraph about the aim and objectives of this work could be redrafted in the introduction part.

Response 2: An additional paragraph about the aim and objectives of this work was added.

Comments 3:  If possible, please provide the information about the latest data of the economic importance of roses from AIPH in the introduction part.

Response 3: We agree. The following sentence was added to the revised version of the manuscript. According to The International Association of Horticultural Producers (AIPH), the total number of roses exported to EU-countries in 2020 was about 8.06 billion flowers, contributing to a total value of $1.4 billion (AIPH, 2021)

Comments 4: There are many different types of soilless substrate using in cut roses, please give a brief introduction. To my knowledge, cocopeat is also widely used in cut roses, please add some more information in the introduction part.

Response 4: We agree. We added the types of soilless substrates used in cut rose to the third paragraph of the introduction section.

Comments 5: The ratio of tuff and cocopeat could be added at the first look, Line 75-76.

Response 5: We agree. text revised.

Comments 6:  Line 128. there is no necessary to compare May and Sep.

Response 6: We agree. Sentence removed.

Comments 7:  Line 152-153, i do not understand how you get the result. Please explain the part clearer. Response 7: We agree. Text revised.

Comments 8: Line180-181, Line185-186. is it a assumption or you proved it?

Response 8: The sentence in line 180-181 is assumption. We removed it from the revised version. However, the sentence in 185-186 is a finding by previous study.

Comments 9: In 2.2, the index of Photosynthetic need to change to photosynthetic rate (Pn).

Response 9: The index (Pn) was removed from the introduction part. Now photosynthesis rates are consistent across the manuscript.

Comments 10: The replicate numbers (n=?) should be added to each table.

Response 10: replicate numbers added to each table.

Comments 11:  The meaning of testing the indicator of “flower diameter after five days” needs to be explained, why you choose this indicator for postharvest quality?

Response 11: Thank you for pointing this out. "Flower diameter after five days" is the measurement of the flower's widest point five days after it has been cut and placed in a vase. The indicator "flower diameter after five days" was chosen to evaluate the long-term effect of treatments on the flower's physical characteristics. This measurement helps evaluate the lasting impact on the flower’s health and appearance, provides insights into post-treatment effects, simulates real-world conditions important for marketability, and ensures consistency in the experimental methodology. However, the term ‘Flower diameter after five days’ was replaced with ‘Flower diameter after five days in vase’. The definition of the term was added to the methodology.

Comments 12:   Please check the language carefully, for example, Line 190, tough or though? Figure 3 is not clear, please improve the resolution.

Response 12: We Agree. tough in Line 190 is wrong. The word was revised in the text.  For Figure 3, we increased the figures' resolution and added more details to Figure 3 to increase its clarity.

Reviewer 3 Report

Comments and Suggestions for Authors

1) What is meaning in practice that 0-4 and 0-8 mm particles? It should be calssified into subgroups according size and give the percentage of each group.

2) If there is relative large percentage of small particles (such as 0.001-0.005mm), do they loss  (by irrigation water) during the planting?

3) Fig. 1. where are the closed and open symbols in ther figure?

4) Fig. 2. How did the data get?

5) Fig. 3 and 4. The figures are so unclear.

6) Line167. please show the data of pH and EC during the planting. 

Author Response

Dear Reviewer,

Thank you for your constructive and thoughtful comments. We have revised the manuscript according to your feedback and suggestions. As detailed below, we have addressed the concerns you raised. We believe these revisions have significantly strengthened the manuscript and hope you will now find it suitable for publication.

Comments 1:  What is meaning in practice that 0-4 and 0-8 mm particles? It should be calssified into subgroups according size and give the percentage of each group.

Response 1: We appreciate your suggestion to classify particles into more specific subgroups and provide percentages for each group. Based on the specific characteristics and requirements of our research objectives, we decided to divide particles into three size ranges: 2-4 mm, 0-4 mm, and 0-8 mm. The distinctions between these size ranges allow us to precisely analyze the effects and behaviors of particles within these predefined categories. While we recognize the value of further subgrouping and providing detailed percentages, we believe that the current approach effectively meets the goals of our study without compromising the clarity and relevance of our findings. We hope that this explanation addresses your concerns about the particle size classification in our manuscript.

Comments 2:  If there is relative large percentage of small particles (such as 0.001-0.005mm), do they loss  (by irrigation water) during the planting?

Response 2: We used 100-micron nonwoven geotextile membrane to hold the growing substrates and to prevent fine particles from immigrating to the drainage layer. After washing the media for the first time, the drained water was clear. Moreover, a filter was placed before the fertigation tank and we did not notice any sediments on the filter.  However, geotextile membrane size was added to the text.

Comments 3: Fig. 1. where are the closed and open symbols in the figure?

Response 3: Good point. The caption of Figure 1 is revised.

Comments 4:  Fig. 2. How did the data get?

Response 4: We agree. We added that to the text. ‘The weather data inside the greenhouse during the experiment was recorded by Hobo Pendant temp/light data logger (Onset Computer Corp., Bourne, MA).’

Comments 5:  Fig. 3 and 4. The figures are so unclear.

Response 5: Sure. We increased the figures' resolution and more details were added to Figure 3 and Figure 4 to increase its clarity.

Comments 6:  Line167. please show the data of pH and EC during the planting.

Response 6: Thank you for pointing this out. We measured the pH and EC for the growing substrates at the beginning of the experiment. During the experiment, we measured the drained water in the collector tank weekly to maintain control over the fertigation solution (data not recorded). Text was added to the methodology to explain the measurement time and meters. For Line 167, clarification was added too.

‘The Electrical Conductivity meter (YK-22CT, Lutron Electronic Enterprise CO, Taipei, Taiwan) and pH meter (pp-203 pH, GOnDO Electronic Co, Taipei, Taiwan) were measured at the beginning of the experiment for all growing substrates (Table 1)’

Reviewer 4 Report

Comments and Suggestions for Authors

Dear Authors,

After reviewing your manuscript entitled “Evaluating the Performance of Cocopeat and Volcanic Tuff in Soilless Cultivation of Roses,” I have identified some points that require clarification and improvement, particularly concerning the statistical analysis.

1. It is essential to clearly indicate whether the values presented in the tables are accompanied by the standard error of means (SEM) or standard deviation of means (SD). This distinction is critical for readers to correctly interpret the variability and reliability of your data. Please revise the tables to specify whether SEM or SD is used.

2. The manuscript mentions the use of the Least Significant Difference (LSD) test for post hoc analysis. However, the LSD method does not provide full control over the experiment-wise type I error rate, potentially leading to an increased likelihood of false positives. The Tukey Honest Significant Difference (HSD) test is generally preferred in such cases as it offers better control over type I error rates. Please consider using the Tukey HSD test for post hoc comparisons or provide a rationale for the choice of the LSD test, addressing its limitations and justifying its appropriateness for your analysis.

3. Several figures in the manuscript are not clear, which hampers the ability of readers to fully understand and interpret your results. Please provide high-resolution versions of the figures to ensure that all details are visible, and the figures are of publication quality.

4. There are several grammatical and structural errors in the manuscript, please carefully read the manuscript, and fix them all.

Comments on the Quality of English Language

Moderate English Language editings are required. 

Author Response

Dear Reviewer,

Thank you for your constructive and thoughtful comments. We have revised the manuscript according to your feedback and suggestions. As detailed below, we have addressed the concerns you raised. We believe these revisions have significantly strengthened the manuscript and hope you will now find it suitable for publication.

After reviewing your manuscript entitled “Evaluating the Performance of Cocopeat and Volcanic Tuff in Soilless Cultivation of Roses,” I have identified some points that require clarification and improvement, particularly concerning the statistical analysis.

Comments 1:  It is essential to clearly indicate whether the values presented in the tables are accompanied by the standard error of means (SEM) or standard deviation of means (SD). This distinction is critical for readers to correctly interpret the variability and reliability of your data. Please revise the tables to specify whether SEM or SD is used.

Response 1: We agree. The figure caption was revised and standard error of means (SEM) was added to the text.

Comments 2: The manuscript mentions the use of the Least Significant Difference (LSD) test for post hoc analysis. However, the LSD method does not provide full control over the experiment-wise type I error rate, potentially leading to an increased likelihood of false positives. The Tukey Honest Significant Difference (HSD) test is generally preferred in such cases as it offers better control over type I error rates. Please consider using the Tukey HSD test for post hoc comparisons or provide a rationale for the choice of the LSD test, addressing its limitations and justifying its appropriateness for your analysis.

Response 2: We agree. In response to your concern regarding the use of the Least Significant Difference (LSD) test for post hoc analysis. To address this issue, we have conducted a reanalysis of all data using the Tukey Honest Significant Difference (HSD) test.

Comments 3: Several figures in the manuscript are not clear, which hampers the ability of readers to fully understand and interpret your results. Please provide high-resolution versions of the figures to ensure that all details are visible, and the figures are of publication quality.

Response 3: Sure. We increased the figures' resolution and more details were added to Figure 3 to increase its clarity.

Comments 4:  There are several grammatical and structural errors in the manuscript, please carefully read the manuscript, and fix them all.

Response 4: Thank you for your valuable feedback. We thoroughly reviewed the manuscript with the assistance of a colleague to improve the language and correct any grammatical and structural errors. We appreciate your review and are dedicated to ensuring the highest quality in our work.

Round 2

Reviewer 2 Report

Comments and Suggestions for Authors

Well revised. It can be accepted.

Comments on the Quality of English Language

Fine.

Author Response

Dear Reviewer,

Thank you for your positive feedback and accepting our revised submission. We appreciate your thorough review and constructive feedback, which significantly helped us improve the quality of our work.

Best Regards

Reviewer 3 Report

Comments and Suggestions for Authors

For the particle size. (1) 0-4 mm includes 2-4mm; (2) we know 0-4 mm means particle size is < 4mm, but it is too wide range. for example, the fraction of  <1mm is 90%, 50% or 10%? It is very important. (3) it is the same question for 0-8 mm. 

Author Response

Comments 1: For the particle size. (1) 0-4 mm includes 2-4mm; (2) we know 0-4 mm means particle size is < 4mm, but it is too wide range. for example, the fraction of  <1mm is 90%, 50% or 10%? It is very important. (3) it is the same question for 0-8 mm.

Response 1: We agree with this comment. Therefore, we analyzed the distribution of particle size for tuff 2 to 4 mm, tuff 0 to 4 mm, and tuff 0 to 8 mm that were used in the tested treatments. A new figure has been added to the manuscript (Figure 5) to show the distribution percentages for each type.

Reviewer 4 Report

Comments and Suggestions for Authors

The authors have addressed all the reviewer's comments, and the paper is now in a publishable form.

Comments on the Quality of English Language

Minor English Language Editings are needed.

Author Response

Dear Reviewer,

Thank you for your positive feedback and for reviewing our revisions. We appreciate your time and input in improving our manuscript.

Best regards

Round 3

Reviewer 3 Report

Comments and Suggestions for Authors

All information is produced now.